# Long-Lasting, Fine-Tuned Anti-Tumor Activity of Recombinant *Listeria monocytogenes* Vaccine Is Controlled by Pyroptosis and Necroptosis Regulatory and Effector Molecules

**DOI:** 10.3390/pathogens13100828

**Published:** 2024-09-25

**Authors:** Abolaji S. Olagunju, Andrew V. D. Sardinha, Gustavo P. Amarante-Mendes

**Affiliations:** 1Departamento de Imunologia, Instituto de Ciências Biomédicas, Universidade de São Paulo, São Paulo 05508-000, SP, Brazil; abolajisamson@icb.usp.br (A.S.O.); andrewsardinha@icb.usp.br (A.V.D.S.); 2Instituto de Investigação em Imunologia, Instituto Nacional de Ciência e Tecnologia (INCT-iii), São Paulo 05508-000, SP, Brazil

**Keywords:** *L. monocytogenes*, anti-tumor vaccine, CD8 T cell, pyroptosis, necroptosis

## Abstract

One of the main objectives of developing new anti-cancer vaccine strategies is to effectively induce CD8+ T cell-mediated anti-tumor immunity. Live recombinant vectors, notably *Listeria monocytogenes*, have been shown to elicit a robust in vivo CD8+ T-cell response in preclinical settings. Significantly, it has been demonstrated that *Listeria* induces inflammatory/immunogenic cell death mechanisms such as pyroptosis and necroptosis in immune cells that favorably control immunological responses. Therefore, we postulated that the host’s response to *Listeria*-based vectors and the subsequent induction of CD8+ T cell-mediated immunity would be compromised by the lack of regulatory or effector molecules involved in pyroptosis or necroptosis. To test our hypothesis, we used recombinant *L. monocytogenes* carrying the ovalbumin gene (LM.OVA) to vaccinate wild-type (WT), *caspase-1/11*^−/−^, *gsdmd*^−/−^, *ripk3*^−/−^, and *mlkl*^−/−^ C57Bl/6 mice. We performed an in vivo cytotoxicity assay to assess the efficacy of OVA-specific CD8+ T lymphocytes in eliminating target cells in wild-type and genetically deficient backgrounds. Furthermore, we evaluated the specific anti-tumor immune response in mice inoculated with the B16F0 and B16F0.OVA melanoma cell lines. Our findings demonstrated that while caspase-1/11 and GSDMD deficiencies interfere with the rapid control of LM.OVA infection, neither of the KOs seems to contribute to the early activation of OVA-specific CTL responses. In contrast, the individual deficiency of each one of these proteins positively impacts the generation of long-lasting effector CD8+ T cells.

## 1. Introduction

Cancer vaccines and immunotherapies have emerged as revolutionary approaches to cancer treatment, offering new hope and improved patient outcomes [1,2,3,4]. However, patient response rates vary across different types of cancer and depend on patients’ individual characteristics [5]. Notably, numerous types of cancers are poorly immunogenic, and the tumor microenvironment (TME) is often immunosuppressive and/or refractile to the infiltration of immune cells [6]. To overcome such obstacles, efforts are underway to improve tumor immunogenicity and change the nature of the TME to an immune-active state [6,7].

An innovative and efficient method for developing a novel vaccine formulation is to use live bacteria as a vector to deliver heterologous (tumor-associated) antigens. This offers a unique approach to enhancing immunogenicity in the context of cancer immunotherapy [8]. Currently, 16 active or recruiting clinical trials employ bacteria to treat cancer [9], with countless others completed in the last two decades. Several use *Listeria monocytogenes* (LM) as a therapeutic platform because of their unique characteristics and ability to strongly induce both cellular and humoral immune responses [10,11,12,13] against infectious diseases and cancer due to the properties of intracellular transmission [14]. 

These engineered vectors not only deliver tumor antigens to dendritic cells but can also offer intrinsic adjuvant capacity since they activate multiple pattern recognition receptors (PRRs) present in immune cells [10,15]. These PRRs activate signaling pathways responsible for upregulating proinflammatory genes like type-I Interferons (IFNs), Tumor necrosis factor alpha (TNFα), Interleukin (IL) 8, and pro-IL-1β directly. In addition, PRRs control the fate of infected or surrounded cells by their interplay with the host cell death molecular machinery [16]. For instance, cytosolic PRRs may assemble multimolecular structures called *Inflammasomes*, which initiate caspase-1-dependent inflammatory cell death (pyroptosis) [17,18,19]. Inflammasomes are important in innate immunity to pathogens, but their role in modulating adaptive immunity is still not fully understood [19]. Interestingly enough, inflammasome activation was shown to limit the generation of antigen-specific T-cell response to *L. monocytogenes* [20,21].

Necroptosis is another inflammatory cell death process that can be triggered by PRRs, such as Toll-like receptors (TLRs) 3 and 4, DNA-dependent activator of IFN-regulatory factors (DAI), Retinoic acid-Inducible Gene I (RIG-I), and Melanoma Differentiation-Associated protein 5 (MDA-5) [16,22]. Necroptosis is essentially controlled by the Receptor-interacting protein kinase 3 (RIPK3) phosphorylation of mixed lineage kinase domain-like pseudokinase (MLKL), which in turn migrates to the cell membrane, forming lytic pores. Remarkably, although the RIPK3-MLKL axis was shown to be activated by *L. monocytogenes* and to contribute to an efficient host defense mechanism, it seems that the phosphorylation of MLKL induced by *L. monocytogenes* infection did not lead to the death of the host cell [23].

Overall, it is not clear how pyroptosis- or necroptosis-related genes participate in the immune response triggered by *L. monocytogenes*, particularly in the context of vector-delivering tumor-associated genes. Therefore, this study aimed to assess the ability of live recombinant *L. monocytogenes* to stimulate CD8+ T cell-mediated anti-tumor immunity in animal models lacking the essential regulatory and effector molecules of necroptosis (RIPK3 and MLKL) and pyroptosis (caspase 1/11 and Gasdermin D (GSDMD)).

## 2. Materials and Methods

### 2.1. Mice 

C57BL/6 *ripk3^−/−^*, *casp-1/11^−/−^*, and *gsdmd^−/−^* mice were generously provided by Vishva Dixit (Genentech, Inc., South San Francisco, CA, USA), Richard Flavell (Yale University, New Haven, CT, USA), and Petr Broz (University of Lausanne, Lausanne, Switzerland), respectively. *mlkl^−/−^* mice were obtained from Douglas Green (St. Jude Children’s Research Hospital, Memphis, TN, USA) and used under the terms of the material transfer agreement obtained from James Murphy (The Walter and Eliza Hall Institute of Medical Research-WEHI, Melbourne, Australia). Six- to eight-week-old WT, *casp-1/11^−/−^*, *gsdmd^−/−^*, *ripk3^−/−^*, and *mlkl^−/−^* mice were used as control and experimental groups. All mouse experiments were performed in the animal facilities of the Institute of Biomedical Sciences, University of São Paulo, under the guidelines of the Ethics Committee on Animal Use, University of São Paulo. 

### 2.2. Recombinant Listeria Monocytogenes Culture and Infection

The recombinant *LM* strain (10403S) expressing OVA was described previously by Dudani and collaborators [24]. For infections/vaccinations, frozen stocks of *LM*-OVA were thawed, serially diluted in PBS, and used to intravenously infect/vaccinate mice via the retro-orbital plexus at 1 × 10^3^ CFU in 100 µL of PBS or otherwise indicated. 

### 2.3. Bacterial Burden per Spleen 

Spleens of all infected mice were harvested on day 3 or 7 post-infection, and single-cell suspensions were prepared in an RPMI-1640 medium (Life Technologies, Burlington, ON, Canada). CFU/spleen was determined by plating 10-fold serial dilutions of single-cell suspensions on BHI-Streptomycin agar plates. 

### 2.4. In Vivo Evaluation of CD8 T-Cell Effector Activity

An evaluation of in vivo antigen-specific CTL killing was accomplished as previously described by Clemente and collaborators [25]. Briefly, spleens from WT donor mice were harvested and processed for single-cell suspensions by tweezing the organs in a cell strainer with a syringe plunger. Cells were counted, divided equally into two populations, and marked separately with low (1 µM) or high (10 µM) concentrations of carboxyfluorescein succinimidyl ester (CFSE). CFSE^High^ cells were pulsed with 10 nM of OVA_257-264_ (SIINFEKL) peptide, while the control CFSE^Low^ remained unpulsed. The two populations of cells were washed and mixed in a 1:1 ratio. A total of 2 × 10^7^ cells in 100 µL of non-supplemented RPMI-1640 were injected into the *LM*-OVA-vaccinated and control mice via the retro-orbital sinus, at 7 or 27 days post-infection. After 16–20 h, spleens from recipient mice were excised, processed, and submitted to flow cytometry (BD CANTO, BD, Mountain View, CA, USA). Data were analyzed using the FlowJo v10 workspace. 

The percentage of specific lysis was determined using the following formula: $$\frac{1-\%\;{\mathrm{CFSE}}^{\mathrm{High}}\;\mathrm{vaccinated}\;\;\mathrm{group}/\%\;{\mathrm{CFSE}}^{\mathrm{low}}\;\mathrm{vaccinated}\;\mathrm{group}\;\times 100}{\%\;{CFSE}^{\mathrm{High}}\;\mathrm{control}\;\mathrm{group}/\%\;{\mathrm{CFSE}}^{\mathrm{low}}\;\mathrm{control}\;\mathrm{group}}$$

### 2.5. In Vivo Tumor Growth

To address the anti-tumor potential of *LM*-OVA, mice were vaccinated or not, and 7 or 27 days later, they were inoculated with 1 × 10^6^ B16F0 and 1 × 10^6^ B16F0.OVA melanoma cells in each right and left flank, respectively. Tumor diameters were measured using a caliper at 2-day intervals using the formula below.
**V = 0.5 a × b^2^**
a = long diameter of the tumor; b = short diameter of the tumor.

Mice were euthanized, and tumors were excised when the tumor diameter reached about 1 cm^3^.

### 2.6. Statistical Analysis

Statistical analysis was conducted using GraphPad Prism version 8 (GraphPad Software Company, Inc., Boston, MA, USA). An ordinary one-way ANOVA or two-way ANOVA was used to determine statistical significance, followed by Bonferroni post-tests. Statistical differences were considered significant when the *p* value was less than 0.05.

## 3. Results

### 3.1. Pyroptosis- but Not Necroptosis-Related Proteins Affect LM-OVA Bacterial Burden

We first evaluated the ability of our genetically modified mice to control the LM-OVA bacterial burden as a measure of the efficiency of the immune response triggered by our recombinant vaccine vector. At the same time, we examined the safety of our vaccine strategy in each genetic background. On day 3, the number of colony-forming units (CFU) found in the spleens of casp.1/11^−/−^ and gsdmd^−/−^ mice was higher compared to that found in WT, ripk3^−/−^, and mlkl^−/−^ mice. This suggests that pyroptosis- but not necroptosis-related proteins are important for controlling the early stages of LM-OVA infection in our experimental conditions (Figure 1A). Importantly, the higher bacterial burden did not result in any overt adverse effect in these animals. It is worth noting that on day 7 post-infection, mice from all deficient backgrounds resolved the infection, except for casp.1/11^−/−^ mice that still showed a low but significant level of bacterial burden in their spleens (Figure 1B). Our findings indicate that the LM-OVA vaccine is safe in all deficient genetic backgrounds and that the caspase 1/11-GSDMD axis participates in the clearance of LM-OVA infection. These results suggest that the deficiency of these pyroptosis-related proteins may impact the immune responses to the recombinant vaccine antigen (OVA), thereby weakening the LM-OVA vaccine’s potential in individuals displaying such defects. Therefore, we designed experiments to address the ability of LM-OVA to induce antigen-specific CTL response in WT and genetically deficient mice.

### 3.2. Deficiency of Pyroptosis- or Necroptosis-Related Proteins Did Not Impair Early Antigen-Specific CTL Response Triggered by LM-OVA Vaccination 

To test the possibility that LM-OVA would induce inferior OVA-specific CD8+ T cell-mediated immune responses when presented in pyroptosis- or necroptosis-deficient backgrounds, we measured the in vivo elimination of OVA_257-264_ peptide-pulsed target cells in WT and knockout mice vaccinated or not with LM-OVA using the in vivo cytotoxicity assay as previously described [25]. 

On day 7 post-vaccination, the peak of CD8 T-cell response, we observed a virtually complete elimination of OVA peptide-pulsed target cells from the spleens of both WT and knockout mice (Figure 2), suggesting that the individual deficiency of casp.1/11^−/−^, gsdmd^−/−^, ripk3^−/−^, or mlkl^−/−^ did not blight the early antigen-specific CTL effector response triggered by LM-OVA vaccination. 

To further support these observations, we asked whether the absence of either of these proteins would hamper anti-tumor immune responses triggered by LM-OVA. The B16F0 and B16F0.OVA melanoma cell lines were inoculated subcutaneously in the right or left flanks of WT and knockout mice on day 7 post-LM-OVA vaccination, and tumor growth was monitored every other day (Figure 3A). Remarkably, while B16F0 cells formed tumors in all LM-OVA-vaccinated animals, no B16F0.OVA tumors were observed in either WT or knockout mice, suggesting that the LM-OVA vaccine conferred protective anti-tumor activity regardless of the genetic deficiency of pyroptosis- or necroptosis-related proteins (Figure 3B). Our results also confirmed that the anti-tumor effect of the LM-OVA vaccination is antigen (OVA)-specific. It is important to mention that the in vivo elimination of target cells completely depends on CD8 T lymphocytes, as shown by the lack of an anti-tumor effect of LM-OVA-vaccinated cd8^−/−^ mice (Appendix A). Also, importantly, B16F0 and B16F0.OVA tumors displayed similar growth in all unvaccinated groups, as expected (Figure 3B). 

### 3.3. Late Antigen-Specific CTL Response Triggered by LM-OVA Vaccination Is Weakened in Pyroptosis- and Necroptosis-Deficient Mice 

To verify whether the efficiency of LM-OVA vaccination was long-lasting, we repeated the in vivo cytotoxic assay at 27 days post-vaccination and found a significant decrease in the elimination of the target cells in all mouse strains, suggesting that our vaccination protocol is insufficient to induce long-lasting effector and/or memory antigen-specific CD8+ T cells (Figure 4). Most importantly, deficiency in either pyroptosis- or necroptosis-related proteins marginally aggravated the loss of LM-OVA-induced CTL activity over time (Figure 4).

To analyze whether the declining CTL response also impacted the anti-tumor effect of LM-OVA vaccination, we inoculated B16F0 and B16F0.OVA melanoma cells after 27 days of vaccination and measured tumor growth every two days. Similar to our previous results, we observed a decline in the efficiency of the LM-OVA vaccination to protect from B16F0.OVA melanoma growth in all mouse strains (Figure 5A). Importantly, to our surprise, casp.1/11^−/−^, gsdmd^−/−^, ripk3^−/−^, and mlkl^−/−^ mice exhibited slightly enhanced protection compared to WT mice.

Finally, we tried to circumvent this late-time inefficiency by increasing the dose of the LM-OVA vaccine 5-fold. Remarkably, all mice showed complete protection, as none of the B16F0.OVA tumors inoculated 27 days after high-dose LM-OVA vaccination grew in this condition (Figure 5B), much like the mice inoculated with B16F0.OVA 7 days after low-dose vaccination (Figure 3). Taken together, our results indicate that pyroptosis- and necroptosis-related molecules fine-tuned the efficiency of LM-OVA vaccination, predominantly with respect to the generation of long-lasting effector and/or memory antigen-specific CD8+ T cells.

## 4. Discussion

Vaccines have provided novel prospects for preventing and treating numerous diseases, including cancer [26]. Recent research has demonstrated that immunotherapy based on live, attenuated, and/or recombinant bacteria offers advantages in increasing the immune response to immunosuppressed malignancies. *Listeria monocytogenes (LM)* has demonstrated powerful innate and adaptive immunity in numerous scenarios, making it a leading candidate for therapeutic bacteria [26]. Indeed, there are several *LM*-based vaccines used as therapeutic vaccines for cancer immunotherapy against cervical [27], prostate [28], pancreatic, lung, ovarian, mesothelioma [29], and colon cancer [30], among others, showing promising results in clinical trials.

The impact of cell death on the intricacies of adaptive immunity has been extensively investigated for decades. Regardless, the precise mechanisms by which diverse cell death pathways influence immune responses during infection remain enigmatic, particularly concerning pyroptosis and necroptosis [21]. We set off trying to find out how well *L. monocytogenes* triggers adaptive immune responses in hosts deficient in key regulatory or effector molecules of necroptosis and pyroptosis. By measuring the bacteria burden in CASP.1/11-, GSDMD-, RIPK3-, and MLKL-deficient mice, we observed that the CASP-1/GSDMD axis participates in the control of *LM*-OVA infection, but the deficiency in either protein did not impose a lethal susceptibility on the mice. In comparison, the RIPK3/MLKL axis seems to be less important. This is in accordance with the studies by the authors of [31,32], which also reported that the lack of caspase-1/11 in mice makes them more vulnerable to *LM* infection during the early stages of infection (days 3–7). Since caspase-1/11 processes the proinflammatory cytokines IL-1 and IL-18, leading to their release and contributing to the innate immune response and host defense, we consider that IL-1 and IL-18 might also play a role in this process, particularly because *gsdmd^−/−^* mice did not phenocopy the sensitivity displayed by *casp.1/11^−/−^* mice. We are currently investigating the role of IL-1/IL-18 signaling in LM-OVA-inducing CD8+ T-cell responses. 

Importantly, the induction of adaptive immunity requires live, replicating bacteria, as killed or inactivated vaccines generally do not induce optimal protective immunity [33,34]. The notion that *L. monocytogenes* elicits T cell-mediated immunity has led many researchers to consider it a promising and effective recombinant vaccine vector for the stimulation of cell-mediated immunity [35,36,37]. Several preclinical studies have demonstrated that listeria and listeria-based vaccines can effectively induce potent anti-tumoral immunity against various tumor-specific antigens [38,39,40,41]. 

In our study, we evaluated the ability of *LM*-OVA to induce an in vivo antigen-specific anti-tumor CD8+ T-cell response in wild-type and necroptosis- or pyroptosis-deficient mice. Several studies have established that antigen-specific CD8 T-cell responses peaked 7–8 days after vaccination/infection in cases of intravenous *LM* infection. In our study, we also observed a strong induction of in vivo antigen-specific CD8+ T cells on day 7, leading to an almost complete clearance of OVA-expressing target cells in all the mouse strains. In addition, our study assessed the prophylactic efficacy of *LM*-OVA vaccination in WT and KO mice subcutaneously inoculated with the B16F0 and B16F0.OVA melanoma cell lines on day 7. *LM*-OVA completely inhibited the growth of OVA-carrying tumors (B16F0.OVA) in WT and KO mice, showing that, at the peak of the response, the induction of antigen-specific CD8+ T cells is successfully translated to an anti-tumor therapy irrespective of the host deficiency in either pyroptosis or necroptosis machineries. This anti-tumor activity shown by *LM*-OVA is completely CD8-dependent (as shown in Appendix A).

Studies have demonstrated that memory CD8 T cells last for extended periods following infection or vaccination. However, sustaining memory CD8 T cells is an ongoing and ever-changing process. Memory CD8 T cells experience temporal alterations in their characteristics and abilities. Consequently, even among a group of memory CD8 T cells that target a single antigen, significant diversity exists in their physical attributes and functional capabilities. The *LM* model has been extensively employed in many research studies to investigate and elucidate the mechanism(s) that regulate alterations in pathogen-specific CD8 T cells during memory development [42]. In relation to this, we decided to explore the ability of LM-OVA to induce antigen-specific memory CD8+ T cells in our model.

In comparison to day 7, we observed that the elimination of target cells was relatively inefficient on day 27, suggesting that our *LM*-OVA vaccination protocol could not provide an effective memory immune response. Similarly, modest protection against tumor growth was observed at this later time point. Interestingly, each deficient mouse strain displayed a minor yet significantly enhanced protection against B16.OVA tumor growth compared with WT mice (Appendix A), suggesting that increased *LM*-OVA availability due to a deficient genetic background might help in priming a better memory immune response. In line with this, by increasing the vaccine dose (5-fold), we could expand the protective anti-tumor response in all mouse strains. Notwithstanding the differential anti-tumor response observed in our experimental settings, we recognize that we did not formally evaluate the frequency of OVA-specific memory T cells in individual mouse strains and temporal conditions. Therefore, experiments using MHC class I tetramers are under consideration and should be performed in the near future in our laboratory. Taken together, our results suggest that necroptosis or pyroptosis molecules take part in the long-lasting, fine-tuned CD8+ T cell-mediated anti-tumor activity of the recombinant *Listeria monocytogenes* vaccine, but the deficiency of these molecules can be circumvented by increasing vaccine concentration and perhaps adding a second or third boosting dose. This is also under investigation in our laboratory. 

## 5. Conclusions

In conclusion, the ability of live vaccine vectors to trigger an efficient, protective host immune response is fine-tuned by the host genetic landscape, including genes related to necroptosis and pyroptosis. This accentuates the need for further investigation on the development of customized approaches to improve the effectiveness of vaccination protocols, particularly in immunodeficient patients [13]. 

## Figures and Tables

**Figure 1 pathogens-13-00828-f001:**
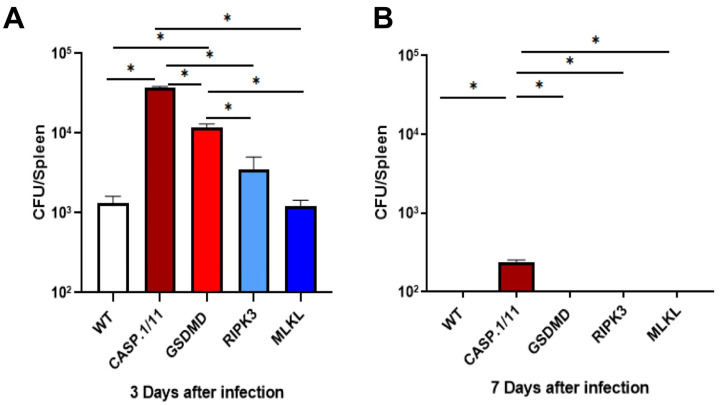
Bacterial burden in LM-OVA-infected mice. After 3 (**A**) and 7 (**B**) days post-LM-OVA (10^3^ CFU) infection, mice were euthanized, and spleens were harvested and processed as described. Results are expressed as mean CFU ± standard deviation per group (n = 5) and represent three independent experiments. One-way ANOVA and Bonferroni post-tests were both used in statistical analysis. * *p* < 0.05.

**Figure 2 pathogens-13-00828-f002:**
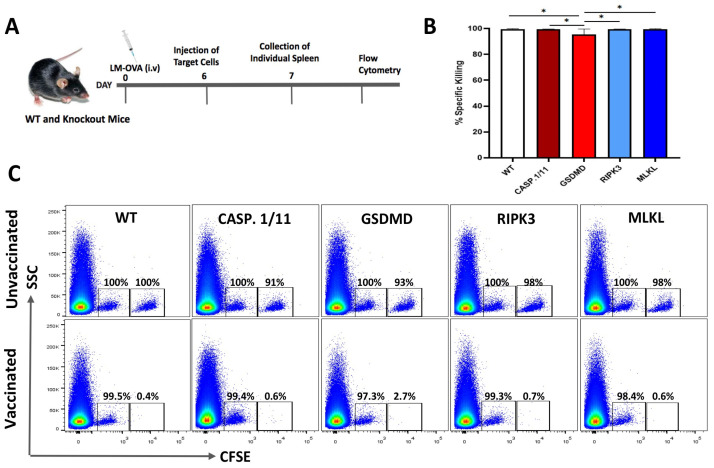
Early in vivo CTL activity triggered by LM-OVA vaccination in WT and necroptosis- or pyroptosis-deficient mice. (**A**) WT and knockout mice were vaccinated or not with LM-OVA and 6 days later injected with a single cell suspension containing OVA-pulsed target cells, as described in Material and Methods. On the next day, mice were euthanized, and spleens were processed for flow cytometry. (**B**) A bar chart showing the percentage of the LM.OVA-induced in vivo elimination of target cells in all mouse strains on day 7. (**C**) Representative flow cytometry density plots. The results are expressed as the mean ± standard deviation per group (n = 5) and represent three independent experiments. A one-way ANOVA and Bonferroni post-tests were both used in the statistical analysis. * *p* < 0.05.

**Figure 3 pathogens-13-00828-f003:**
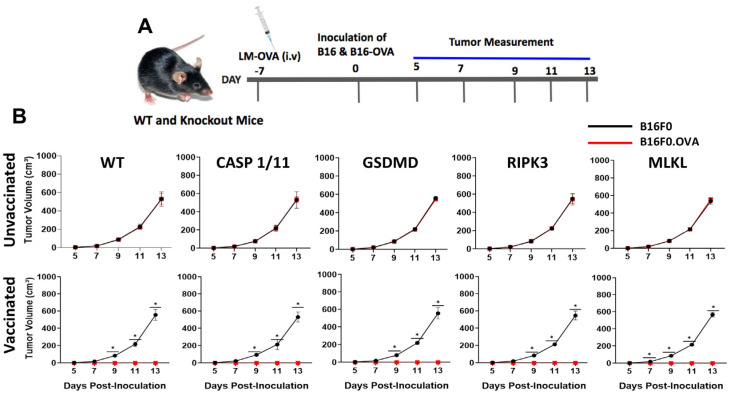
Early anti-tumor immune response induced by LM-OVA in WT and knockout mice. (**A**) Mice were vaccinated or not with LM-OVA and inoculated with the B16F0 and B16F0.OVA melanoma cell lines after 7 days. (**B**) Tumor growth was evaluated every two days. The results are expressed as the mean ± standard deviation per group (n = 5) and represent three independent experiments. A two-way ANOVA and Bonferroni post-tests were both used in the statistical analysis. * *p* < 0.05.

**Figure 4 pathogens-13-00828-f004:**
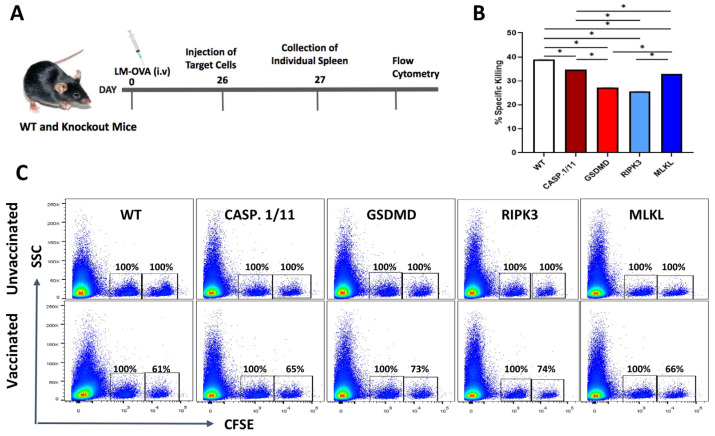
Late in vivo CTL activity triggered by LM-OVA vaccination in WT and necroptosis- or pyroptosis-deficient mice. (**A**) WT and knockout mice were vaccinated or not with LM-OVA and 26 days later injected with a single cell suspension containing OVA-pulsed target cells, as described in M&M. On the next day, mice were euthanized, and spleens were processed for flow cytometry. (**B**) A bar chart showing the percentage of the LM.OVA-induced in vivo elimination of target cells in all mouse strains on day 27. (**C**) Representative flow cytometry density plots. The results are expressed as the mean ± standard deviation per group (n = 5) and represent three independent experiments. A one-way ANOVA and Bonferroni post-tests were both used in the statistical analysis. * *p* < 0.05.

**Figure 5 pathogens-13-00828-f005:**
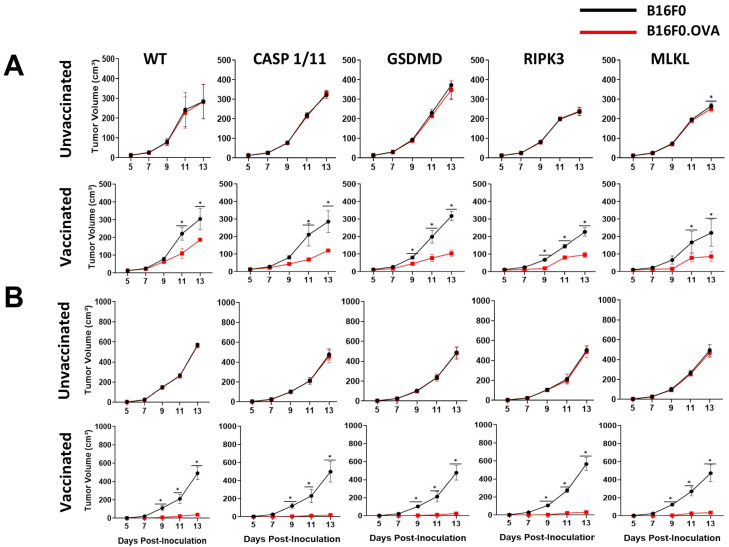
Late anti-tumor immune response induced by LM-OVA in WT and knockout mice. (**A**) Mice were vaccinated or not with LM-OVA (1 × 10^3^) and inoculated with the B16F0 and B16F0.OVA melanoma cell lines after 27 days. Tumor growth was evaluated every two days. Partial protection was observed in all groups. (**B**) Full protection was observed in all groups of mice vaccinated with a high dose of LM-OVA (5 × 10^3^). The results are expressed as the mean ± standard deviation per group (n = 5) and represent three independent experiments. A two-way ANOVA and Bonferroni post-tests were both used in the statistical analysis. * *p* < 0.05.

## Data Availability

Raw data will be provided upon request.

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
