# Peer review of "Long-Lasting, Fine-Tuned Anti-Tumor Activity of Recombinant Listeria monocytogenes Vaccine Is Controlled by Pyroptosis and Necroptosis Regulatory and Effector Molecules"

_pathogens, 2024, doi:10.3390/pathogens13100828_

Round 1
Reviewer 1 Report
Comments and Suggestions for Authors
The manuscript submitted by Abolaji S. Olagunju, Andrew V. D. Sardinha and Gustavo P. Amarante-Mendes elegantly discusses how genes related to pyroptosis or necroptosis participate in the immune response triggered by L. monocytogenes (Listeria monocytogenes are small, facultatively anaerobic, Gram-positive bacilli that can occur alone or in pairs or short chains. It is a species of bacteria capable of causing diseases in humans, such as meningitis), particularly in the context of tumor-associated genes that deactivate the vector.
The authors have very clearly demonstrated that the ability of live vaccine vectors (using Listeria monocytogenes) to trigger a protective and proven effective immune response in the host can be tuned by the host's genetic landscape, including genes related to necroptosis and pyroptosis. This highlights the need for further research into the development of personalized approaches to improve the efficacy of vaccination protocols, particularly in immunocompromised patients.
The results are very well presented and are widely discussed in the text. The authors are to be congratulated for the excellent manuscript submitted.
In short, the study is interesting and extremely relevant to the area.
Author Response
We would like to thank very much this referee's very positive evaluation of our manuscript!
Reviewer 2 Report
Comments and Suggestions for Authors
The manuscript lacks a detailed mechanistic study, and the presented results are not entirely convincing. More comprehensive experimental data are needed to strengthen the findings.
The authors claim they tested the LM-OVA vaccine using several gene knockout mouse models. However, verification of the gene knockouts via Western blot is essential after the transfer of mice from different institutes.
Figure 1, it is unclear whether the higher bacterial burden results in any adverse effects on the mice. This information should be included.
Figure 2, the CFSE assay is somewhat unclear. The authors should clarify the meaning of the percentage numbers indicated in the plot (e.g., 100% and 91%) and explain how these percentages relate to the gating of specific cell populations.
Figures 3 and 4, it is a bit unusual to observe such uniformly sized tumors with very small SD value. Typically, there is significant variation in tumor size for B16 cells. The authors should provide the original tumor photos to substantiate their findings.
To assess the effectiveness of the OVA vaccine in the different experimental groups, the authors should examine the OVA-specific T cell response using tetramers and analyze their percentage among total CD8+ lymphocytes.
Comments on the Quality of English LanguageNA
Author Response
Thank you very much for taking the time to review this manuscript. Please find the detailed responses enclosed and the corresponding revisions highlighted the re-submitted files.

Reviewer 3 Report
Comments and Suggestions for Authors
The study of Abolaji S. Olagunju and colleagues aimed to assess the ability of the live recombinant Listeria. monocytogenes (LM) to stimulate CD8+ T cell-mediated anti-tumor immunity in animal models lacking the essential regulatory and effector molecules of necroptosis (RIPK3 and MLKL) and pyroptosis (Caspase 1/11 and Gasdermin D). Genetically modified casp.1/11-/- , gsdmd-/-, ripk3-/- and mlkl-/- mice and recombinant vaccine vectors carrying the ovalbumin gene (LM-OVA) were used. First was evaluated the ability of these genetically modified mice to control the LM-OVA bacterial burden as a measure of the efficiency of the immune response triggered by this recombinant vaccine vector. Deficiency of pyroptosis- or necroptosis-related proteins did not impair the early antigen-specific CTL response triggered by LM-OVA vaccination. Then they verified whether the absence of either of these proteins would hamper anti-tumor immune responses triggered by LM-OVA, injecting B16F0 and B16F0.OVA melanoma cell lines in wild type (WT) and knockout mice on day 7 post-LM-OVA vaccination. No B16F0.OVA tumors were observed in either WT or knockout mice, suggesting that LM-OVA vaccine conferred protective anti-tumor activity regardless of the genetic deficiency of pyroptosis- or necroptosis-related proteins. However the efficiency of LM-OVA vaccination was not long-lasting, because after 27 days of vaccination the anti-tumor effect of LM-OVA vaccination on B16F0 and B16F0.OVA melanoma cells was decreased.
The authors conclude that the ability of live vaccine vectors to trigger an efficient, protective host immune response is fine-tuned by the host genetic landscape, including genes related to necroptosis and pyroptosis. Further investigations are needed to improve the effectiveness of vaccination protocols.
The study is interesting, but further investigations are needed to deeply evaluate the efficacy of anti-tumor vaccination protocols especially for long-lasting memory.
Some issues should be addressed:
1. .Explain the meaning of RIPK3 and of MLKL
2. Fig. 3 and fig. 5. The font of the words and of the numbers on the graphs is too small and difficult to read. Please, modify.
3. Pag 6 lines 209-211 :” Importantly, once again, casp.1/11-/-, gsdmd-/-, ripk3-/- and mlkl-/- mice exhibited reduced protection compared to WT mice.” I do not think so. Fig. 5A does not show a reduced protection in KO in comparison to WT mice. All groups of mice seem to be partially protected in the same way. Please, explain. I do not understand. Authors do not show any significant difference between the tumor volume of melanoma in vaccinated WT in comparison to each of KO mice. In fact the only significant difference shown is between the tumor volume of B16F0 in comparison to that of B16F0.OVA in the same group of vaccinated mice.
4. DISCUSSION section Pag. 8 lines 288-291. “Similarly, modest protection against tumor growth was observed at this later time point, particularly in each deficient mouse strain, which displayed a minor yet significantly lower protection against B16.OVA tumor growth compared with WT mice.” I do not agree. See what I discussed at point 3.
5. References section: Ref. 26: the name of the journal is repeated twice. Please, correct
Author Response
Thank you very much for taking the time to review this manuscript. Please find the detailed responses enclosed and the corresponding revisions highlightedin the re-submitted files.

Reviewer 4 Report
Comments and Suggestions for Authors
The manuscript "Long-lasting, fine-tuned anti-tumor activity of recombinant Listeria monocytogenes vaccine is controlled by pyroptosis and necroptosis regulatory and effector molecules
'' is focusing on the anti-tumor efficacy of a recombinant Listeria monocytogenes (LM) vaccine. The study investigates how this vaccine, which aims to stimulate CD8+ T cell-mediated immunity, is affected by the presence or absence of specific regulatory molecules involved in cell death processes like pyroptosis and necroptosis. The authors are using various genetically modified mouse models. In their research it is evaluated the vaccine's ability to control tumor growth and the immune response under different genetic conditions. In conclusion the study suggests that while certain genetic deficiencies can alter the early immune response to the LM-OVA vaccine, the overall anti-tumor efficacy is maintained, highlighting the vaccine's potential for broader application even in genetically diverse populations.
General frame
- Deficiencies in pyroptosis-related proteins (caspase-1/11 and GSDMD) impair the rapid control of LM-OVA infection, affecting the early stages of the immune response.
- Necroptosis-related proteins (RIPK3 and MLKL) do not seem to impact the early CD8+ T cell responses significantly.
- Despite some genetic deficiencies, the LM-OVA vaccine was effective in inducing long-lasting anti-tumor immunity, although the absence of these proteins may weaken the overall vaccine potential.
The manuscript has some areas that could be improved:
general text
The manuscript sometimes lacks clear transitions between sections, which makes it hard to follow the progression of ideas. Improving the flow between the sections could make it easier to be read.
Material and method
Some experimental details are not fully described, which could limit the reproducibility of the results. Providing more comprehensive descriptions would benefit the scientific community. Also some of the images are not quite clear.
Statistical analysis
The statistical analysis is not always detailed. A more thorough explanation of the statistical methods used and the justification is neded.
All of the above may enhance quality and impact of the manuscript.
Author Response
Thank you very much for taking the time to review this manuscript. Please find the detailed responses enclosed and the corresponding revisions highlighted in the re-submitted files.

Round 2
Reviewer 2 Report
Comments and Suggestions for Authors
The authors did not fully address my concerns.
Comments on the Quality of English Languagena
Author Response
Again, I would like to thank very much the referee for taking time to evaluate our manuscript and for all suggestions, most of it taken into account to improve the quality of our work.
Unfortunately, at this point we are unable to develop the new experiment using MHC tetramers as suggested, as it would take very long time to import this reagent to Brazil.
We hope that all modification that we made in our manuscript, according to the other corrections suggested by this referee and the other three referees is enough to grant our work publication in Pathogens.
Thank you very much